# Metabolomics-Driven Mining of Metabolite Resources: Applications and Prospects for Improving Vegetable Crops

**DOI:** 10.3390/ijms232012062

**Published:** 2022-10-11

**Authors:** Dhananjaya Pratap Singh, Mansi Singh Bisen, Renu Shukla, Ratna Prabha, Sudarshan Maurya, Yesaru S. Reddy, Prabhakar Mohan Singh, Nagendra Rai, Tribhuwan Chaubey, Krishna Kumar Chaturvedi, Sudhir Srivastava, Mohammad Samir Farooqi, Vijai Kumar Gupta, Birinchi K. Sarma, Anil Rai, Tusar Kanti Behera

**Affiliations:** 1ICAR-Indian Institute of Vegetable Research, Jakhini, Shahanshahpur, Varanasi 221305, India; 2Indian Council of Agricultural Research (ICAR), Krishi Bhawan, Dr. Rajendra Prasad Road, New Delhi 110001, India; 3ICAR-Indian Agricultural Statistics Research Institute, Centre for Agricultural Bioinformatics, Library Avenue, Pusa, New Delhi 110012, India; 4Biorefining and Advanced Materials Research Centre, Scotland’s Rural College, Kings Buildings, West Mains Road, Edinburgh EH9 3JG, UK; 5Department of Mycology and Plant Pathology, Institute of Agricultural Sciences, Banaras Hindu University, Varanasi 221005, India

**Keywords:** metabolomics, metabolome, metabolites, mass spectrometry, NMR, metabolite analysis, vegetables, crop improvement

## Abstract

Vegetable crops possess a prominent nutri-metabolite pool that not only contributes to the crop performance in the fields, but also offers nutritional security for humans. In the pursuit of identifying, quantifying and functionally characterizing the cellular metabolome pool, biomolecule separation technologies, data acquisition platforms, chemical libraries, bioinformatics tools, databases and visualization techniques have come to play significant role. High-throughput metabolomics unravels structurally diverse nutrition-rich metabolites and their entangled interactions in vegetable plants. It has helped to link identified phytometabolites with unique phenotypic traits, nutri-functional characters, defense mechanisms and crop productivity. In this study, we explore mining diverse metabolites, localizing cellular metabolic pathways, classifying functional biomolecules and establishing linkages between metabolic fluxes and genomic regulations, using comprehensive metabolomics deciphers of the plant’s performance in the environment. We discuss exemplary reports covering the implications of metabolomics, addressing metabolic changes in vegetable plants during crop domestication, stage-dependent growth, fruit development, nutri-metabolic capabilities, climatic impacts, plant-microbe-pest interactions and anthropogenic activities. Efforts leading to identify biomarker metabolites, candidate proteins and the genes responsible for plant health, defense mechanisms and nutri-rich crop produce are documented. With the insights on metabolite-QTL (mQTL) driven genetic architecture, molecular breeding in vegetable crops can be revolutionized for developing better nutritional capabilities, improved tolerance against diseases/pests and enhanced climate resilience in plants.

## 1. Introduction 

Metabolites, especially the small molecules, are functional readouts of cellular reactions and are essential for biological processes in all life forms on the earth. They reflect visible traits, characters, and marked changes correlated with the phenotypic parameters of the organisms and maintain them under diverse environmental and pathophysiological conditions. The emergence of organism specific highly transitory and diverse chemical inventories of >200,000 primary and secondary compounds are linked with particular biological activities under adverse environmental conditions [1]. Their importance is reflected in plant-based foods that are reported to contain almost 25,000 different metabolites, of which nearly 7000 are volatile compounds [2]. Whether it may be plant and human cells, or those of unseen tiny microorganisms, it all represents vast cell factories that produce a huge array of metabolites to perform their biological functions [3]. Even a single cell is a biological factory in which thousands of chemical reactions are going on at a given point of time, leading to the biosynthesis of numerous intermediates and the end products. Many of these small molecules have a very short life but help in producing the majority of stable molecules that are the key players for biological functionalities meant for growth and development, reproduction, bioenergetics, defense against stresses and the sustenance of life. The products of biological reactions in the cells, as represented by various metabolic networks, shape the phenotypic characteristics of the organisms [4] and changes in the content or composition of metabolites are reflected by the characteristic traits [5]. 

In the present time, when malnutrition in the human population is continuously increasing worldwide, vegetable crops have emerged as a prominent source of nutritional power with a high content of mineral nutrients, vitamins, proteins, amino acids, carbohydrates, fibers, phenolics, flavonoids and other secondary metabolites [6]. Growing vegetables has not only provided nutritional security [7] but also livelihood options to the masses of people in rural areas [8]. These all-season crops have remained worth-proven for sustaining food systems and enhancing economic self-reliance in the rural population, especially in village women and youth in the tough times of the COVID-19 pandemic, and have supported families of farming communities worldwide [9]. As crop plants, vegetables have a tremendous diversity of secondary chemical molecules [10], which confer valued functions to these commercial agricultural commodities [11]. Functional phytometabolites derived from vegetable sources have been considered a nutritional powerhouse for human health [12]. Therefore, exploring a vast majority of vegetable plants, among as many as 10,000 plant species defined as vegetables [13] for their metabolite profile, is a profound task with multiple benefits. The cutting-edge technologically advanced metabolomics platforms for metabolite identification, functional characterization, library generation, comparative analysis and data resources have made the exploration of vegetable bioactive metabolites an easy and less time-consuming operation. 

## 2. Vegetable Plants as Prominent Metabolite Resources

In the human diet, vegetables are integral constituents to enrich defense-activating, immunity-boosting and health promoting biomolecules [14]. Carotenoids, polyphenolics, flavonoids, sterols, phenolic acids, glucosinolates, glycosides and dietary fibers from vegetables are the risk-reducing factors against diseases and stresses [15]. Phytometabolites, having bioactive antimutagenic, antimicrobial, antiviral, antidiuretic, antidiabetic, cytotoxic and antioxidant mechanisms, have been isolated and identified from lettuce, asparagus, broccoli, spinach, tomato, chili, mustard, turnip, Brussel sprouts and cauliflower [16]. The content and bioactivity level of active molecules in the tissues of vegetable plants depends on crop species, cultivars, agronomic practices, agro-climatic impacts, and post-harvest management [17]. Since the consumption of health foods with high nutraceutical values are majorly linked with prevention and avoidance of major degenerative and deadly diseases, vegetables and fruits with functionally diverse bioactive metabolites are the obvious choice in the diet of the society. High throughput methods have made the comprehensive identification and functional characterization of metabolites from vegetable plants more efficient, easy, integrative and cost-effective [18].

The complex evolutionary diversification pattern of crop plants is reflected in their metabolic diversity [19]. Diversified bioactive functions associated with the chemical diversity in vegetables provide commercial edge to these crop commodities [11]. Tomatoes with rich antioxidant potential are among the most nutritious crops having multi-minerals, proteins, essential amino acids, carbohydrates, vitamins, fatty acids, phytosterols, carotenoids, polyphenolics and flavonoids as reactive oxygen species (ROS) scavengers and reducers of oxidative damage [20]. Growth stage-dependent differential metabolic regulation of free, bound and conjugated polyphenolic metabolite signatures was observed in different tissue-types of tomato, eggplant and pepper [21]. Caffeic, ferulic, chlorogenic and *p*-coumaric acids and flavonoids naringenin, catechin, kaempeferol, quercetin and rutin in tomato varieties and their commercial by-products confer antioxidant properties [22]. Other phenolics like benzoic, 4-hydroxybenzoic and syringic acids impart varied defense roles to tomato against biotic interactions [23]. Marked changes happened in polyphenolics content of ripening tomato fruits cultivated under organic and conventional agronomic practices [24]. 

Eggplants (*Solanum melongena*) hold high nutritive values due to vitamins, polyphenolics, flavonoids and anthocyanins as free-radical and superoxide scavenging antioxidant biomolecules [25]. Although excessive domestication caused a reduction in phenolic constituents in eggplant [26], metabolic constituents like carbohydrates, vitamins, ascorbic acid, organic acids etc., are largely cultivation dependent [27]. From the fruits of eggplant, malonated caffeoylquinic acid and 5-O-caffeoylquinic acid derivatives [28] and from the peels, antioxidant nasunin, a derivative of delphinidin anthocyanin [29], delphinidin-3-rutinoside and delphinidin-3-rutinoside-5-glucoside [30], delphinidin-3-(caffeoylrutinoside)-5-glucoside and petunidin-3-(*p*-coumaroylrutinoside)-5-glucoside [31] were isolated and identified. These bioactive compounds contribute to the antioxidant properties, color, morpho-physiological traits and the phenotypic characters of eggplant. Fruits of the *Capsicum* species (pepper) are a good source of secondary metabolites quercetin 3,7-di-O-α-l-rhamnopyranoside and narigenin-7-O-β-d-(3-p-coumaroyl)-glucopyranoside [32]. The high content of carbohydrates, organic acids (ascorbic, citric, malic, quinic, succinic, fumaric and oxalic acids), capsaicinoids (capsaicin, dihydrocapsaicin, nordihydrocapsaicin, homocapsaicin, homodihydrocapsaicin), carotenoids, flavonoids, luteolin and vitamins (E, C and A) that add to the antimicrobial, antiseptic, anticancer, counterirritant, appetite stimulator, immunomodulator and antioxidant properties [33] were reported from the fruits of *Capsicum annuum* L., *C. chinense* Jacq. and *C. baccatum* L. [34]. The world’s hottest chili from the northeast region of India, called Naga King (*Capsicum chinense* Jacq), possesses unique pungent capsaicinoid in red fruits, along with amino acids, palmitic, stearic and α-linolenic acid and a high content of linoleic acid in the seeds [35]. Phenotypic functions of mature chili fruits were influenced by the content of ascorbic acid, capsaicinoids and flavone aglycone [36] and color intensity traits are impacted by carotenoid biosynthesis and sequestration [37]. Among the most important causal conditions for secondary chemical diversification in chili are the ecological succession and adaptive crop domestication that benefit plants against stresses.

Brassica plants are rich in flavonol-O-glycosides quercetin, kaempeferol and isorhamnetin, anthocyanins pelargonidin, cyanidin, delphinidin, peonidin, ptunidin and malvidin and phenolic acids hydroxycinnamic, sinapic, *p*-coumaric and ferulic acids [38]. Brassicaceae vegetables (cabbage, cauliflower, broccoli, radish, cherry radish, watercress and Brussel sprouts) biosynthesize volatile organic compounds (VOCs) from various metabolic pathways like fatty acid metabolism, methylerythritol phosphate (MEP) pathway and glucosinolate (GLS) [39]. From *Brassica rapa*, 38 bioactive metabolites including tocopherols, sterols, carotenoids, glucosinolates and policosanols [40], and from nine other Brassicaceae plants 28 metabolites have been identified [41]. In Cucurbitaceae plants (cucumbers, gourds and melons) cucurbitacins A, B, C, D and E [42], along with the phytoalexin C-glycosyl flavonoid [43] and other antioxidant and free radical scavenging compounds, have been identified. Additionally, a significant shift in metabolite content was observed during the developmental stages of stylar, intermediate and peduncular segments in cucumber plants [44]. Overall, vegetable plants, especially the fruits, are a rich pool of secondary metabolites having diversified metabolic origin and multipronged functional properties, which not only benefit the host plants but become beneficial for the human beings too.

## 3. Defining Metabolites, Metabolome and Metabolomics

Plant metabolites are the ultimate bio-molecular products of multiple pharmaco-physiological and biochemical functions that characteristically represent phenological traits of the organisms. These diverse groups of phytometabolites are regularly being produced within cells in response to the climatic perturbations, to represent strategic tolerance and survival in organisms. The plant-specialized primary and secondary compounds help organisms to regulate their energy metabolism, photosynthesis and respiration, signal recognition, totipotency and defense mechanisms to improve their fitness under biotic challenges and adverse environments. Therefore, the real time identification of the cellular metabolites and characterization of their metabolic networks and biosynthetic routes becomes essential. It is also pertinent to link metabolites and their functional attributes with the proteins and genes responsible for their biosynthesis in metabolic biosynthetic pathways. 

Metabolome represents a qualitative and quantitative collection of an entire set of small molecule low-molecular weight metabolites essential for cell growth, maintenance and normal biological functions [44]. Boththe composition of biogenic VOCs, the volatilome [45] and non-volatile metabolites constitute global metabolome of the organisms [46]. Catalyzed by the enzymes regulated by the proteomes, the metabolites are synthesized in the cells in continuum, but are subject to change in a time dependent manner to determine the metabolic functions of the organisms [47]. Active cell metabolome affects cellular physiology, which further modulates various omics-level changes that affect genomic, epi-genomic, transcripomic and proteomic levels [48]. The dynamics of chemical complexity in cellularly processes is reflected in the biological functions and [49] the genetic makeup of the organisms [48].

Metabolome composition can be induced by the environmental stressors (light, humidity, temperature, salt, drought or water-excess), biotic factors (pathogens/pests/beneficial microorganisms), the exogenous application of chemicals and the contamination of soil, water and fertilizers [50]. A global unbiased extraction, separation, identification and functional characterization of a whole set of metabolites from cells is crucial [51,52]. Modern methods of ultrasonic extraction [53], membrane separation and molecular distillation [54] have facilitated the extraction of metabolite pools of plant cells. Bioassay (bioactivity)-guided chromatographic separation techniques involving thin layer chromatography (TLC), high performance thin layer chromatography (HPTLC) and high-performance liquid chromatography (HPLC) have widely been employed in the process standardization, method development, detection and the characterization of bioactive metabolites [55].

The technique of studying structural identity, quantity and the characters of metabolites within the whole metabolome is ‘metabolomics’ [44]. The metabolomic interventions offer opportunities to fingerprint key chemical footprints of the cells with high sensitivity, accuracy and preciseness [56,57,58] either in a targeted way [44] or in nontargeted (unbiased) manner [59] (Figure 1). Being viewed as complementary to transcriptomics and proteomics [60], plant metabolomics has versatile implications in metabolome profiling, metabolic pathway dissection, population genetic interpretation, species discrimination and comprehensive analysis [61]. To unravel plant single cell metabolism by exploring single-cell-metabolome, state-of-the-art spatially resolved metabolomics technologies are extremely helpful [62]. The direct relevance and application of the discipline connects with the experiments in biology, nutrition, stress physiology, analytical chemistry, biochemistry, medical sciences, genetics, plant breeding, plant and animal pathology, biotechnology, diagnostics, microbiology and food science [63,64] in a comprehensive workflow (Figure 2).

Comparative metabolomics has become a widely applied and robust tool in cross-boundary applications [65] to discover active drivers of the biological processes under stress conditions. Qualitative and quantitative metabolomics is helpful to understand fundamental behavior of plant phenotypic responses in tripartite interactions in relation to the development, physiology, tissue identity, resistance and tolerance against biotic and abiotic factors [66]. With stringent unsupervised statistical methods, principal component analysis (PCA) approaches [67] and bioinformatics tools, the study discloses complex, intricate and interactive nature of metabolic networks associated with the inherent metabolic changes in biological systems [68] (Figure 3). Tripartite interactions in plants display the most natural ping-pong mechanisms guided by intrinsic chemical factories through which the cellular machinery overcomes the impact of environmental stressors [50].

Gene expression and proteomic analyses do not reflect a cumulative status of biochemical happenings within a cell [69]. Additionally, the studies on individual enzymes of the metabolic pathways, the means by which enzyme networks control metabolite concentrations and fluxes within the cells in an integrated manner remains less understood [70]. Likewise, because of the limitations, proteome at a given point of time may not be completely predicted from the transcriptome due to specific differences in regulatory mechanisms [71]. The study of global metabolome is therefore down the line from the gene and protein functions, and reflects more closely the ever-changing activities and functional mechanisms within the cells at a given point of time [44,72].

## 4. Application of Metabolomics in Vegetable Crops

The comprehensive metabolomic profiling of vegetable crops has yielded large datasets of metabolites, complex biological pathways and hidden regulatory networks linked with distinguished phenotypic traits [73]. The mapping of different classes of metabolites in vegetable plants, localizing distinct cellular metabolic pathways, classifying the functional performance of metabolite molecules and establishing linkages between metabolic fluxes and genomic and proteomic regulations, can yield targeted results through comprehensive metabolomic experiments that address the plant’s performance in the environment [74]. The majority of work excerpts are presented in Table 1. As key drivers of phenotypic expressions in the organisms, metabolites are essential biomarkers for selective metabolic-quantitative trait loci (mQTL), which links metabolomes with the genomes [57]. Information on metabolite identity and metabolic fluxes in the cells was used to reveal connections between physiological conditions under stresses and proteins/genes to advance metabolomics-driven breeding programs by utilizing mQTLs and metabolome wide genome association studies (mGWAS) [75,76]. These insights can revolutionize molecular breeding in vegetable plants for better nutritional capabilities, improved adoption against diseases/pests and enhanced climate resilience in crops [77].

### 4.1. Vegetable Crop Domestication Studies

Crop domestication processes meant for selecting plants artificially to improve their suitability for human food have profound impacts on plant interactions with their natural inhabitant pests and predators [78]. Since the inception of the domestication processes, the majority of vegetable crops have undergone alterations in their metabolome profile because of the selection pressure [79]. The shaping of the crop metabolism during the domestication processes can be marked at the metabolome level to identify new targets for varietal development through designer metabolic engineering [80]. Tomato has undergone worldwide domestication from wild species through intensive breeding and revealed the evolutionary consequences of improvement to yield high value crop for human consumption and commercial implication [81]. Metabolomic studies clubbed with genomic and transcriptomic analysis indicated that the intensive breeding has changed the global metabolome of tomato fruits [82]. An analysis of natural variation in domesticated tomato (*S*. *lycopersicum*) and wild species (*S*. *pimpinellifolium*, *S*. *peruvianum*, *S*. *habrochaites*, S. *pennellii*, S. *cheesmaniae*, S. *chmielewskii* and S. *neorickii*) using GC-MS and LC-MS revealed pathway structure and metabolic regulation of polyphenolics in fruits and leaves [83]. The selection of the genes associated with the larger fruit size for breeding purposes has changed the associated metabolome profile of the fruits. Making tomatoes phenotypically pink modifies hundreds of metabolites in fruits [82]. The introgression of resistance genes from the wild species has made unexpected alterations in the metabolic profile and was found linked to the fruit-mass altered metabolites [84]. Thus, metabolomics can trace out the manipulated targets of metabolic variations linked to phenotypic and genotypic characters in tomato and other vegetable plants undergoing domestication processes.

### 4.2. Monitoring Development-Dependent Metabolic Changes 

Metabolomic studies on the developmental stages (vegetative, flowering or fruiting) of plants, organs (tissue-dependent growth) or fruits (unripened or have registered marked metabolic shifts linked with varieties, agronomic factors and growth conditions [85,86] (Table 1). The primary metabolite profiling of tomato fruits by combined GC- and LC-MS platforms identified 70 compounds including organic acids, sugars, sugar alcohols, phosphorylated and lipophilic compounds and amino acids [87]. Dry and imbibed tomato seeds analyzed under different development stages showed increased metabolic flux as analyzed by GC-TOF/MS [88]. Quality variables in tomato varieties were analyzed using NMR to enhance economic value to the crop [89]. Light intensity can modulate biochemical pathways and metabolic changes in tomato to impact fruit ripening. LC-MS and GC-MS based metabolomics recorded 554 compounds and revealed that the compounds of health benefits like carotenoids, flavonoids, tocopherols and phenolic acids in the fruits accumulated faster under white light as compared to darkness [90]. A blue light exposure of fruits can effectively alter fruit metabolome during the ripening process and shape the levels of key metabolites imparting fruit quality characters. Early post-anthesis chili fruits accumulated luteolin, apigenin, quercetin, shikimic acid, aminobutyric acid and putrescine as dominant molecules, while chemical dominance at later stages was replaced by leucine, isoleucine, phenylalanine, proline, capsaicin, dihydrocapsaicin and kaempferol [91]. Metabolite profiling by UHPLC-Q-TOF-MS based nontargeted metabolomics assessed distinctly marked varietal variation in chili [92]. The exploration of ethylene-related pathways and carbon metabolism in *Capsicum,* using GC-MS and LC-MS elucidated stage-dependent potential ripening markers and unique ripening-responsible metabolic mechanisms [93]. Metabolomic analyses emphasized that the profile of starch, sugars and sugar-derivatives not only changed following ripening stages but further influenced glycolysis intermediates, tricarboxylic acid cycle, amino acids, pungency precursors and color to also observe significant metabolic shift during non-climacteric ripening of *Capsicum*. 

The skin color of cucumber fruits as determined by chlorophyll and anthocyanin content holds important commercial value for the crop. Comparative metabolomics of dark green vs. light green skinned cucumber deciphered 162 metabolites comprising 40 flavones, nine flavonoids, eight flavanones, eight flavanols, six anthocyanins and other compounds linked with the metabolic pathways imparting skin color [94]. Like cucumber, the sponge gourd *Luffa aegyptiaca* is also a fruit of choice for nutritional purposes among a large population. Metabolic differentiation between the soft young and fibrous mature ripened fruits suggested a complex biochemical process driving different stages of fruit maturation. SPME-GC-MS metabolomic analysis revealed 53 volatile metabolites comprising aldehydes, ketones, acids and aromatic compounds in various proportion while characteristically predominantly present semi-polar compounds oleanane triperpene glycosides and dihydroxy-oxo-oleanonic acid glycosides were determined by UPLC-MS profiling [95]. Floral specialized secondary metabolites, usually pigments and volatiles comprising hydroxycinnamates, phenolamides and flavonoids, are widely produced by the plants to perform a number of physiological functions, including protection against light and ultra-violet radiation, insect invasion, disease-causing organisms and herbivory and pollinator attraction [96]. Metabolomic analysis of 20 different Brassicaceae genotypes using LC-MS revealed 228 detected metabolites and deciphered chemical signatures indicating evolutionary role in floral metabolism in self-compatible and self-incompatible genotypes of Brassicaceae species [97]. Overall, metabolomic studies on vegetable crops comprising different stage-dependent changes in phenotype-guiding chemical constituents like pigments, photosynthates and volatile organic molecules, have led to decipher physiology-driven metabolic patterns in vegetable crops. 

### 4.3. Nutritional Metabolomics of Vegetable Plants

Plant nutrition aspects are essential issues not only for offering quality and economic value to crops but also for providing nutritional security to the people. An accumulation of various minerals and metabolites makes plants nutritionally enriched and secure. Cellular environment drives plant growth, yield and stress resistance/tolerance and offers nutritional value to the crop plants [98]. Metabolic composition, in which small molecule metabolites play a vital role, is crucial for the compositional determinants of the crops for human beings [99]. Nutri-metabolite constituents that add to the nutritional value to the crops are qualitatively and quantitatively determined, analyzed and interpreted to assess vegetables and fruits. Nutri-metabolomics has emerged as a potential analytical approach to investigate the role of vegetables, fruits and plant-derived bioactive compounds, extracts and functional foods on health [100]. The approach involves separation technologies, analytical bioorganic chemistry, precision instrumental analysis, high-throughput instrumentation, chemometrics, biostatistics, databases, complex compound libraries and bioinformatics [97]. At the process level, the sample preparation of biological fluids, the classification of diverse biological stages, the origin and quality of sample, analytical procedures, data acquisition, collection and retrieval, system integration, interfacing, statistical analysis, documentation and the interpretation of the data are the key methodological components in metabolomics [101,102] (Figure 2). The technique leads to the qualitative and quantitative cataloguing of small molecule metabolites of nutritional significance in the plants under environmental pressure, toxicogenic exposure and dietary changes. It could generate a snapshot of metabolome-driven novel gene and protein elucidation in plant species [103] and clubbed with the transcriptomics, can have applications in genome editing studies as well [104]. 

Nutritional metabolomics offers investigations on the application of plant-derived small molecule metabolite profiling to address the complexity of the diet, nutrition, metabolome and health issues [105]. Both the targeted and non-targeted metabolomics of vegetables like tomato [106] and bitter melon (*Momordica charantia*) [107] has implications of advancing nutritional crops [108] by identifying and quantifying nutri-chemicals directly linked with the human diet and nutrition [109]. Comprehensive metabolomic profiling of ripened fruit collection of 32 chilli accessions using nontargeted LC-MS and GC-MS platforms identified 88 semi-polar metabolites. The high accumulation of acyclic diterpenoid phytatetraene and other volatiles were linked with differences in the pungency level of the accessions. Abundant acyclic diterpenoid glycoside capsianosides showed antioxidant and pharmaceutical properties in *C. annuum* [110]. The linking of specific metabolites or metabolic pathways directly to a trait of nutritive value that can potentiate human health is being achieved through interspecies comparative metabolomics, mQTL and mGWAS [64]. This could yield identification of candidate genes for the important traits in tomato [111]. Breeding crops with high content of mineral nutrients, health promoting metabolites and high-value nutraceuticals is now becoming easier due to advancements in comprehensive crop metabolomics [82,112]. Although chemical composition traits are yet to become established markers of the crop quality, research on protein, oil, provitamin A content in maize, starch content in potato and rice and carotenoids in tomato [113] has advanced our understanding of these traits. 

Worldwide, the work on metabolomics-inspired data generation registering metabolic changes on the impacts of environmental and genetic perturbations in plants has gained interest due to their influence on nutrition in human diet and commercial value of plant-derived products [114]. NMR helped to screen spoilage of tomato fruits during storage [115] to decipher impact low- and high-lycopene content in tomato sauces on human beings [116] and to differentiate 361 samples of tomato plants grown under organic and conventional cultivation practices [117]. A metabolomics approach using QTOF-MS revealed that tomato grown under an organic agronomic environment differentially expressed antioxidant activities and the commercial ketchup product made from the fruits accumulated significantly high antioxidant micronutrients [118]. Non-targeted metabolomics helped metabolic fingerprinting of organically and conventionally grown tomato and pepper crops for differential food markers [119,120]. A comprehensive metabolomics approach clubbed with genomic, transcriptomic and proteomic work can address crop nutritional quality parameters for engaging the same in vegetable improvement programs [121]. Comparative metabolomics addressing food and nutritional values in vegetables varieties being grown under different agroecological conditions or cultural practices (organic vs. non-organic) could identify specific small molecule signatures for improving nutrition-specific agricultural production [122].

### 4.4. Decephering Plant’s Adaptive Strategies under Abiotic Stresses

In fields, vegetable crops are directly exposed to abiotic stresses including drought, salinity, heat, cold, radiation and edaphic factors. All these factors negatively modulate plant performance in terms of physiological, biochemical and molecular changes at the cellular levels [123]. Intrinsically, a plant’s adaptive mechanisms and mitigation strategies are linked with modifications in biological metabolic processes like photosynthesis, energy metabolism, hormone synthesis, secondary metabolite biosynthesis and the quantitative accumulation of metabolites following stresses [124]. Metabolome analysis can address a plant’s adaptive changes at metabolic level. Cold-susceptible tomato fruits revealed seven amino acids, 16 sugars, 27 organic acids and various other metabolites due to chilling shock [125]. Besides the intrinsic capability of crop genotypes to fight against stresses, other factors also influence plant responses against abiotic stresses. Seed priming agents influence the metabolic profile of the crop plants to make them more stringent and tough. Tomato seed priming with β-aminobutyric acid (BABA) triggered induced tolerance against environmental stress in plants [126] and identified 6887 metabolite features of interest in MS-generated metabolomics analysis [127]. Various transcription factors including *TDR4* gene played a crucial regulatory role in the ripening of tomato fruits. Metabolomics analysis together with RNA-seq has led to the revelation that *TDR4* silenced tomatine accumulation, while tyrosine, phenylalanine and organic acids were reduced in tomato [128]. Phenomics investigations coupled with MS-based metabolomics of bio stimulant 4-VITA treated tomato plants showed the effective elicitation of secondary metabolic pathways including the biosynthesis of phytohormones, thylakoid lipids, xanthins and chlorophyll [129]. A coordinated reprogramming of biosynthetic pathways enhances tomato plant resilience against drought stress. LC-MS based untargeted metabolomic experiment unraveled modulations in the biochemical framework of tomato plants following non-microbial silicon (Si)-based bio stimulant application under saline conditions [130]. The primary metabolic pathways of tricarboxylic acid cycle, fatty acid and amino acid biosynthesis along with the secondary metabolic phenylpropanoid and flavonoid biosynthetic pathways, including flavone and flavanol synthesis were reprogrammed in tomato leaves due to Si-based bio stimulant exposure to support plants in alleviating salinity stress. The exogenous application of ascorbic acid improved spinach plant tolerance against freezing conditions. Leaf metabolomic analysis revealed greater abundance of antioxidants (glutathione, tocopherols), compatible solutes (proline, galactinol and myo-inositol) and activated enzyme activity [131]. Abiotic stress mitigation strategies in vegetable plants essentially comprise elicitation of metabolite-based crop responses followed by changes in proteomic and transcriptomic levels. Metabolomics can detect and identify new metabolite of interest, the abundance of which relates to stress minimization or investigates multi-fold changes in the level of diverse metabolites involved in tolerance. 

### 4.5. Assessing Metabolome under Plant-Microbe Interactions 

Metabolomics clubbed with other omics studies integrates multi-layered information to decipher the complexity of dynamic molecular and metabolic changes in plants due to microbial and environmental interactions [50]. The metabolomic analysis of microbial interactions with plants identified biomarkers with immense applications in biosensor development, phytopathogen monitoring, disease diagnostics and overall plant protection [132]. Vegetable plants continuously interact with microorganisms that majorly influence their cellular metabolic composition and affect their yield performance in field conditions [133]. Beneficial microbial interactions not only promote growth and development of plants but also improve intrinsic tolerance/resistance leading to plant immunity [134]. Non-targeted UHPLC-MS and targeted UHPLC-triple quadrupole-MS metabolomics analysis of tomato primed with *Pseudomonas*
*fluorescens* N04 and *Paenibacillus alvei* T22 documented crucial metabolic changes in response to the infection of *Phytophthora capsici*. Tomato root treatment with two PGPR strains, followed by the inoculation of the phytopathogen *P. capsici* at the stem, caused differential reprogramming of aromatic amino acids, glycoalkaloids, benzoates, phenylpropanoids, flavonoids, organic acids, oxygenated fatty acids and abscisic and methyl salicylic acid as major signal molecules in plant tissues [135]. 

Non-targeted metabolomics of bacterial pathogen-challenged tomato identified activation of phenylpropanoid pathway with the increased accumulation of quinic acid and flavonoid derivatives as antimicrobial agents and hexoses as osmotic regulators [136]. Bacterial wilt caused by *Ralstonia *solanacearum** created the blockage of tomato xylem vessels due to the accumulation of putrescine, as revealed by LC-MS analysis [137]. The response of two near-isogenic lines of tomato showing different levels of resistance against white-fly mediated begomovirus tomato curly stunt virus (ToCSV) was documented using UPLC-MS based metabolomic analysis. The reprogramming of structurally diverse metabolites in susceptible plants has led to identify biomarkers such as hydroxycinnamic acid conjugated to quinic acid, galactaric acid and glucose [138]. Molecular and biochemical mechanisms underlying quantitative resistance in two advanced diploid potato genotypes were investigated using metabolome profiling [139]. Metabolites belonging to phenylpropanoids, flavonoid and alkaloid chemical groups were highly induced in resistant genotypes than in susceptible plants. From expression studies it was concluded that the resistance to late blight in resistant genotypes was mainly associated with cell wall thickening due to the deposition of hydroxycinnamic acid amides and the accumulation of flavonoids and alkaloids.

GC-MS and LC-MS metabolomics directly linked primary and secondary metabolic status of resistance and susceptible plants with specialized defense responses against bacterial, fungal or viral pathogens [140]. NMR has been universally accepted to identify plant metabolome under the integrated pest management (IPM) framework and to address candidate secondary compounds involved in host-pest interactions and resistance [141]. Investigations on cultivated and wild tomato genotypes having resistance against wilt and blight diseases [142], *Fusarium oxysporum* [143], pest (*Frankliniella occidentalis*, western flower thrips) [141,144] and root-knot nematode [145] using the NMR metabolomic platform, reflected the chemotaxonomic perturbations of resistance and/or susceptibility. Distinct metabolic profile using ^1^H NMR was observed in chili under beet mild curly top virus (BMCTV) infection. Infected plants showed high abundance of marker molecules fructose, isoleucine, histidine, tryptophan and phenylalanine linked to the energy metabolism for viral acquisition, replication and capsid assembly [146]. Plant interactions with pathogenic microorganisms result in programmed modulation in cellular machinery that initiates damaging impacts on crop quality and yield [147]. Under pathogenic conditions, plants utilize pathogen-triggered defense mechanisms to modulate composite cellular metabolome in response to patho-physiological conditions [148]. Mapping structurally complex metabolite cascade in plant cells after pathogenic attack and determining interactive metabolic outcomes allows for the identification of those signature molecules, which directly or indirectly influence progression or suppression of pathogens [149] and paves the way to identify signature molecules.

### 4.6. Metabolomic Studies in Plant-Insect (Herbivore) Interactions 

Under insect interactions, a plant’s metabolome undergoes metabolic changes due to induced cellular mechanisms involved in the interplay of herbivorous relationships and their implications at the biological interface [150]. Pest infestation on crops usually favors activation of defense metabolites to check pest invasion, develop resistance against insect attack and facilitate physiological growth [151]. Characteristic cellular metabolomic signatures confined to these processes can be deciphered in local or systemic parts through targeted or nontargeted analysis. The specific impacts of various trophic levels on the primary and secondary metabolic profile of tomato plants revealed a unique pattern of non-volatile and volatile metabolites [152]. Metabolome and volatilome analysis of tomato and eggplant challenged with pinworm (*Tutta absoluta*) using ultra-performance liquid chromatography–tandem mass spectrometry (UPLC-MS/MS) and headspace solid-phase microextraction with gas chromatography–mass spectrometry (HS-SPME/GC-MS), revealed 797 primary and secondary metabolites and 141 VOCs representing aldehydes, alcohols, sugars, aromatics, amines, terpenoids, ketones, phenolics and olefins [153]. The results indicated that because of the characteristic metabolic contents (both volatile and non-volatile), tomato becomes a more suitable host for the pest *T. absoluta* than eggplants. Such compounds could be considered as signature metabolites to be targeted and validated for improving crop breeding [154]. 

GC-MS based nontargeted metabolomics of pepper (*C. chinense* and *C. annuum*) has shown marked changes in metabolite profile related to pest resistance and susceptibility at different developmental stages [155]. The abundance of monomer and dimer acyclic diterpene glycosides (capsianosides) was shown to link with insect resistance, while sucrose and malonylated flavone glycosides with host susceptibility. Enhanced defense in tomato against aphids after *Trichoderma* priming was linked to metabolome and transcriptome reprogramming [156]. The metabolomic study of two pepper (*C. annuum*) genotypes differentially susceptible to spider mite herbivory revealed infestation induced jasmonic acid (JA) and salicylic acid (SA) signaling and terpenoid and flavonoid synthesis [157]. Stronger defense in pepper plants was observed due to the increased production of volatile and non-volatile metabolites induced after metabolic modulations under mite infestation. Transcriptomic rearrangements in plants due to pest attack often leads to altered phenotypic and metabolic responses. In two cucumber (*C. sativus*) genotypes having spider mite (*Tetranychus urticae*) infestation, terpenoid level increased but cucurbitacin C decreased during herbivore interaction. It indicated regulation of directly or indirectly induced defense responses against herbivory [158]. Cucumber plants showed up- and down regulated accumulation of amino acids, organic acids, sugars, glycosides and lipids along with secondary S-containing metabolites as their stress responses against sulfur-fumigation [159]. Qualitative and quantitative screening of okra (*Abelmoschus esculentus* L.) metabolome using ^1^H high resolution NMR revealed isoleucine, γ-aminobutyrate, glutamine, asparagine, choline, phosphocholine and cinnamic acid following postharvest senescence [160]. Knowing the differential interactive performance of plants against insect pests is crucial to deciphering bioactivity-guided chemical reactions at the host-insect interface. Untargeted metabolomics could be helpful to identify candidate metabolites linked with the host specificity against insect pests.

### 4.7. Understanding on Rhizosphere Metabolome 

In the systems biology approach, rhizosphere is considered as a living biological entity to be studied for rhizodeposition of diverse metabolites, rhizomicrobiome dynamics and plant growth promoting mechanisms. Due to belowground beneficial microbial interactions, rhizosphere continues to influence crop performance under diverse plant growth conditions [161]. Environmental metabolomics can efficiently address complex, dynamic and edaphic factor-influenced rhizosphere metabolomes of vegetable crops. Inoculation of plant growth promoting bacteria (PGPB) in tomato rhizosphere under saline stress caused the accumulation of proline and other osmoprotectants in the root exudate. It has modulated plant responses due to shikimate and phenylpropanoid pathways and differential carbon substrate utilization by the microbial communities [162]. Resistant cultivars of cucumber offer control against Fusarium wilt disease in the field. Nontargeted metabolome profiling of root exudates of susceptible cucumber cultivar and investigations on rhizosphere microflora showed excessive release of organic acids (citric, pyruvate, succinic and fumaric acid) under higher abundance by Foc-susceptible cultivar. Resistant cultivar allowed the recruitment of specific beneficial bacterial communities from Comamonadaceae [163]. Pathogen infection has been shown to change the microbiome and metabolome composition of the plant rhizosphere. The invasion of the phytopathogen *Ralstonia solanacearum* causing global bacterial wilt disease in tomato influenced the abundance of rhizosphere microbial communities. High pathogenic abundance reduced alpha-bacterial diversity, especially those of *Bacillus* and *Chitinophaga* in tomato rhizosphere. Metabolome analysis using GC-MS revealed that low abundance of pathogen harbored more sugar metabolites [164]. Accumulation of *p*-hydroxybenzoic acid (PHBA) in the soil becomes auto toxic for cucumber plants. However, PHBA stress mitigation in cucumber is achieved by inoculating *Klebsiella* in the soil, which not only decomposed the auto toxic compound but also activated soil enzyme activity, enhanced the antioxidant effect in plants and systemically accumulated secondary metabolites in leaves, as revealed by metabolome and transcriptome experiments [165]. This is despite the fact that the role of rhizosphere interactions in offering different levels of disease resistance using metabolomics is less studied. 

Plant genotypes, edaphic factors, symbiotic or pathogenic interaction and pest infestation influence chemical diversity of rhizosphere metabolome and the belowground microbiome composition. GC-TOF-MS based metabolome analysis revealed cultivar, organ and developmental stage-dependent quantitative differences in phytochemical profile [166]. The exudates of tomato fruit, root and shoot as surface compounds supported the growth of *Salmonella enterica*. Short-term and long-term exposure to sulfur fumigation exhibited differential metabolic patterns in cucumber in protected cultivation. Metabolomic profiling of rhizosphere and bulk soils detected 11 different metabolites including organic acids and sugars and pointed out differentially expressed starch and sucrose metabolic pathways [167]. Soil metagenomics reflected differential structural and functional bacterial community composition in pepper rhizosphere, in which metabolites were more abundant than in the bulk soil. Under greenhouse cultivation conditions, a close proximity was shown between the rhizosphere bacterial communities and soil metabolome. Such studies indicated the processes of qualitative and quantitative chemical diversity of surface exudates secreted by the plant organs on bacterial population dynamics. The findings may be manipulated for developing better plant genotypes, with efficient surface secondary metabolites having role in food safety, plant defense and health. 

### 4.8. Metabolomic Studies on Transgenic Vegetable Plants 

The ^1^H NMR-based metabolomics of transgenic tomato fruits engineered to accumulate high content of polyamines, spermine and spermidine revealed accumulated organic acids, amino acids and sugars as compared to non-transgenic plants [168]. The accumulation of nitrogenous spermine and spermidine metabolites in transgenic tomato fruit cells enhanced stimulation of carbon sequestration and improved respiratory activity due to the up-regulation of phosphoenolpyruvate carboxylase. This mechanism has given transgenic plants metabolic advantage over the control non-transgenic plants. In the high temperature regime associated with global climate changes, plants suffering from heat stress may be relieved by the suppression or over-expression of heat shock proteins (HSPs) that control thermotolerance related metabolic functions. Metabolome profiling of heat stress transcription factor (HsfsB1) transgenic tomato plants and wild-type (WT) revealed a higher accumulation of biosynthetic products related to phenylpropanoids and flavonoid pathways, including the isomers of caffeoylquinic acid in transgenic plants due to overexpression of HsfsB1 [169]. A close correlation between the thermo-stress response of tomato plants and their physiological development under heat regime was observed. 

### 4.9. Metabolomic Changes in Vegetable Plants Due to Nanoparticles Impact 

Phytotoxic impact of zinc oxide nanoparticles (ZnO-NPs) due to the exposure of plant roots and crop growth modulatory property of silica-nanoparticles (SiO_2_-NPs) has been widely reported. Cucumber foliage when exposed to ZnO-NPs and SiO_2_-NPs changed plant performance as was reflected by the comparative metabolomic profiling of seedlings in which both the primary (amino acids, organic acids, sugars and glycosides) and secondary metabolites were changed [170]. The study established that instead of root exposure of the NPs, foliar application may become rather more effective in generating positive plant response and showing better crop performance. Silver ions and silver-NPs (AgNPs) induce oxidative stress in cucumber. GC-MS based metabolomics analysis has resulted in the identification and quantification of 268 metabolites in AgNPs-treated cucumber leaves [171]. Metabolic reprogramming as revealed by multivariate analysis showed activation of defense related polyphenolic compounds, pentadecanoic and arachidonic acids whereas down-regulation of linoleic and linolenic acids, glutamine, asparagine and photosynthesis-linked metabolites and has improved understanding on molecular mechanisms of cucumber while responding to AgNP exposure. Metabolomics based on GC-TOF-MS assessed metabolic changes in spinach (*Spinacia oleracea*) leaves exposed to Cu(OH)_2_ nanopesticide (NP) and Cu ion upon foliar spray. Zhao et al. [172] showed activated anti-oxidative pathways but reduced beneficial antioxidant levels of various metabolites including ascorbic acid, alpha-tocopherol, threonic acid, β-sitosterol, 4-hydroxybutyric acid, ferulic acid and phenolics in Cu(OH)_2_-exposed spinach leaves. Cu^2+^ was supposed to reduce the antioxidant levels and confer changes in ascorbate and aldarate metabolism resulting in the reduced nutritional value of Cu-NP exposed spinach plants. Foliar exposure of leaves to CeO_2_ NP has induced down-regulation of various amino acids and marked invisible metabolic reprogramming in spinach [173]. 

### 4.10. Metabolome Profiling and Pharmaco-Physiological Impact Assessment 

Chemotype differentiation in bitter gourd (*Momordica charantia*, Cucurbitaceae) was performed using UPLC-HRMS based metabolomics [174]. Crop fruits with potential anti-diabetic properties accumulated high content of secondary metabolite flavonoids, phenolics, terpenoids and sterols, while terpene metabolites were linked with fruit bitterness. Metabolome diversity in edible vegetable fruits makes them potentially useful against human diseases and exploring composition of known and unknown metabolites can be performed with metabolomics. Phyto-medicinal biomolecules from fruit sub-parts like pericarp, skin and seeds were identified by NMR-based metabolic profiling of bitter gourd [175]. The clustering of metabolites in seeds and the pericarp of fruits was observed using multivariate analysis. However, the metabolome profile of fruit skin revealed accumulation of phytosterols charantin and momordicine. Differential accumulation of secondary metabolites in different fruit parts and phytosterols in the skin was shown to have links with antidiabetic properties [175]. However, looking into the pharmaco-physiological, nutritional, health-driving, crop production and commercial aspects directly linked with the global metabolite profile of vegetables crops, extensive studies are required. 

**Table 1 ijms-23-12062-t001:** MS and NMR based metabolomics applications in vegetable crop plants.

Sr. No.	Vegetable Plants/Parts	Metabolomics Platforms	Metabolome Studies/Metabolites Identified	Reference
**Tomato**				
1.	Transgenic and non-transgenic fruit powder	600 MHz ^1^H-NMR in D_2_O buffer solution	amino acids (9), gama-aminobutyric acid (GABA), organic acids, sugars, choline, polyamines spermine and spermidine	[168]
2.	Ripened fruits of 50 cultivars including cherry and round type tomato	^1^H-NMR, LC-(QTOF) MSIntra-(NMR–NMR, LCMS–LCMS) and inter-(NMR–LCMS) metabolomics-correlation analyses	phenylalanine, zeatin hexose, caffeic acid hexose, dehydrophaseic acid, caffoylquinic acid, esculeoside A, rutin, kaempferol-3-*O*-rutinose, dicaffeoylquinic acid, naringenin chalcone-hexose, lycoperoside A–C, tomatin, tomatoside, tricaffeoylquinic acid, naringenin, naringenin chalcone	[176]
3.	Fruit volatile compounds	Headspace SPME-GC-MS	hexanol, phenylethanol, phenylacetaldehyde, hexanal, heptenel, *β*-Ionone, 6-Methyl-5-hepten-2-one, 2-Methylbutanal, 2-isobutylthiazole, 1-Penten-3-one, eugenol	[177]
4.	Plant-bacterial wilt (*Ralstonia solanacearum*) interaction	^1^H-NMR	amino acids, organic acids, and sugar groups, GABA, ethanolamine, glycine, choline, valine etc.	[178]
5.	Midstem and xylem sap-*Ralstonia solanacearum* interaction	LC-MS, GC-MS	amino acids, organic acids, fatty acids, various derivatives of cinnamic acid and benzoic acids, flavonoids and steroidal glycoalkaloids, flavonoids and hydroxycinnamic acids, polyamines and tyramine. 22 compounds identified by GC-MS	[139,140]
6.	Cutin	3D HR-MAS NMR spectroscopy	lignin structures, plant derived fulvic acid, humic materials	[179]
7.	Seeds treated with *Trichoderma* secondary metabolites 6-pentyl-2H-pyran-2-one and harzianic acid and corresponding plants leaves at 15 days	HRMAS NMR	acetylcholine, GABA	[180]
8.	Leaves and roots	GC-MS, LC-MS	sugars, organic acids, oligosaccharides—α-ketoglutarate and raffinose	[181]
9.	Cherry tomato (5 types)—fruits	GC-TOF-MS, UPLC-Linear Trap Quadrupole-Orbitrap-Tandem MS	amino acids, lipids, organic acids, phenylpropanoids, lycopene, β-carotene, and α-carotene	[110]
10.	Mature fruits—25 cultivars	Untargeted MS/MS	7118 accurate mass values	[182]
11.	Fruits—*Cryptococcus laurentii* induced changes	UPLC-MS/MS analysis in Positive and Negative ion mode	59 metabolites phenolics, flavonoids and phenylpropanoids including chlorogenic, caffeic and ferulic acids	[183]
12.	Fruits	GC-MS	primary metabolites (organic acids, sugars, sugar alcohols, amino acids, phosphorylated intermediates, lipophilic compounds)	[87]
13.	Fruits and leaves	^1^H-NMR and LC-QTOF-MS metabolomics profiling	70 metabolites in leaves, 56 in fruits, free amino acids, sugars, sucrose, rhamnose, erythritol, glycerate, alanine, β-alanine, glycine, tyramine, and trigonelline, hydroxycinnamate, flavonoid, or glycoalkaloid, α-tomatine, dehydrotomatine, and chlorogenate	[184]
14.	Fruits *Solanum pannellii*	NMR, HPLC	carotenoids and tocopherols	[185]
15.	Host pathogen interaction with leaf mold disease caused by *Cladosporium fulvum*	GC-TOF-MS, LC-PDA-QTOF-MS	polar primary metabolites (PPM)-132semi-polar secondary metabolites (SPSM)enhanced accumulation of benzoic acid, salicylic acid, tyramine, dopamine, coumaroyltyramine-isoform 1, and coumaroyldopamine	[186]
16.	Predator *Chrysoperla carnea* larvae and herbivore interaction *Tetranychus urticae* and *Myzus persicae*	GC-MS, LC-ESI-MS	non-volatile metabolites—abscisic acid and amino acidsvolatile metabolites—monoterpenes (37%), sesquiterpenes (32%), aldehydes (5%), phenylpropanoids (11%) and fatty acid derivatives (5%)	[152]
17.	Tomato-*Trichoderma harzianum*-Aphids	LC-ESI (+)-MS	primary and secondary semi-polar metabolitesDown-accumulation-citric acid, dihydro-caffeic acid, 2-hydroxyglutarate, phosphoenolpyruvate (PEP), coumarin, syringaldehyde, tetrahydrofolate, δ-tomatine (an alkaloid), N-isovalerylglycine and threonine/homoserine (amino acids) and protoporphyrinogen IX Over-represented metabolites—salicylate β-D-glucose ester and salicyloyl-L-aspartic acid, phenylalanine, coumaric and chorismic acids	[156]
18.	Tomato—thrips-resistant (10 wild) and susceptible (10 cultivated) lines	^1^H-NMR	acylsugars	[187]
19.	Tomato rhizosphere	GC-MS, LC-MS	fatty acid derivatives, oxylipin isomers, azelaic and pimelic acids, glycosides, amino acids	[188]
20.	Tomato leaves—Si-based biostimulant treatment	LC-MS	terpenes, polyamines, phenolic compounds, caffeic-, coumaric-, sinapoyl- and ferulic acids and conjugates, quercetin and kaempferol conjugates, hexadecadienoic acid (HDDA), hydroperoxyoctadecadienoic acid (HPDA)	[130]
21.	Plant-PGPR mediated salt tolerance	GC-MS based metabolite profiling	high accumulation—xylose, rhamnose, D-mannose, L-galactose, allose, sedoheptulose, erythrose, and 2-deoxy-D-erythro-pentitol, lactate and oxalate, alanine, valine, isoleucine, glycine, serine, threonine, and cystathionine in inoculated plants	[189]
22.	Fruits at different ripening stages	FT-ICR MS, NMR, HPLC	amino acids and derivatives, organic acids, sugar esters of caffeic and ferulic acids, caffeoyl- and feruloyl-hexose, monosaccharides fructose, glucose and galactose, choline and adenosine, tannins, flavonoids apigeniflavan, tetrahydroxyflavanone glucoside, polyphenol derivatives like catechin-O-glucoside, catechin-O-rutinoside, dihydrokaempferol, trihydroxy-prenyldihydrochalcone glucosyl-coumarate, quercetin glucoside-glucuronide	[190]
23.	Fruits—chitosan treated plants	LC-ESI-MS/MS	benzenesulfonates, benzoates, desulfoglucosinolates, fatty acids, flavonoids, oligosaccharides, organosulfates, organosulfur compounds, phenylpropanoids, phospholipids and terpenoids	[191]
24.	Tomato product—purees	HPLC-PDA and fluorescence detector, ^1^H-NMR, LC-QToF-MS, LC-multiple reaction monitoring, headspace GC-MS	vitamins, semi-polar and polar metabolites, oxylipins, volatile metabolites, phytonutrients	[192]
**Capsicum**				
25.	*Capsicum annuum* L.—aqueous phase of Serrano peppers	^1^H-NMR, 2D NMR and Chenomx database	vitamin C and 44 metabolites including sugars, organic acids, polyphenols, amino acids, alcohols	[193]
26.	*Capsicum annuum* cv. serrano	^1^H-NMR, 2D NMR and qNMR, PCA, PLS-DS	48 metabolites including sugars (glucose, fructose, sucrose and galactose), organic acids (citric, formic, fumaric, malic), alcohols and polyphenolic acids	[194]
27.	*Capsicum annuum* L. Serrano peppers from two geographically different regions	^1^H-NMR	Veracruz region produce: rich in aspartate citrate, lactate, leucine and sucroseOaxaca region produce: rich in acetate, formate, fumarate, malonate, phosphocholine, pyruvate and succinate	[195]
**Cucumber**				
28.	Cucumber-leaves and root exudates	^1^H-NMR and GC-MS	ascorbic acid, citric acid, amino acids	[196]
29.	Cucumber fruits-extract	^1^H-NMR and GC-MS	nano-Cu influenced metabolite profile of sugars, amino acids, fatty acids, methylnicotinamide, trigonelline, imidazole, quinolate	[128]
30.	Cucumber-fruit peel	UPLC-QTOF-MS	uniquely abundant 113 ions in resistant fruit peels	[197]
31.	Cucumber grafted on different root stocks	GC-MS	volatile organic compounds (VOCs), Fructose-6-phosphate and trehalose), linoleic acid, and amino-acid (isoleucine, proline, and valine)	[198]
32.	*Cucumis sativus*—cucurbit chlorotic yellows virus infection	UPLC, ESI-Q TRAP-MS/MS, MWDB and MRM	total identified metabolites 612. maximum accumulation—lipids, phenolic acids and flavonoidsmetabolites detected—organic acids, alkaloids, terpenoids, lignans, coumarins, and tannins.	[199]
33.	Cucumber fruit-2,4-dichlorophenoxyacetic acid (2,4-D) exogenous treatment	Non-targeted metabolomics by UPLC-qTOF-MS	decreased levels of flavonoids and cinnamic acid derivatives but increased levels of amino acids	[86]
34.	*Luffa aegyptiaca* Mill-young and mature fruits	UHPLC-MS, GC-MS	aldehydes, furans, alcohols, ketones, acids, aromatics	[95]
**Lettuce**				
35.	Lettuce leaf	GC-MS, LC-MS, PCA	101 metabolites from 223 peaks GC-MS, 95 peaks by LC-MS, tetracosanol and hexacosanol, sesquiterpene lactones (lactucopicrin-15-oxalate and 15-deoxylactucin-8-sulfate)	[200]
**Eggplant**				
36.	Eggplant and Tomato—*Tuta absoluta* infection	UPLC-MS/MS, HS-SPME/GC-MS	identified 141 volatile organic compounds and 797 primary/secondary metabolites, dominant compounds-aldehyde, alcohols, alkanes, amine, aromatics, a heterocyclic compound, ketone, olefin, phenol, and terpenes	[154]
37.	Eggplant—fruit accessions	LC-MS, GC-MS	51 compounds (LC-MS), 207 compounds (GC-MS)Terpenes, alkaloids, fatty acids, flavonoids and terpenoids	[201]
38.	Eggplant-fruits-browning mechanism	LC-MS	946 metabolites changed dynamically, 119 common differential metabolites	[202]
39.	Solanaceous crops fruit tissues (peel and pericarp) of pepper, eggplant, tomato, Capsicum annum	LC-ESI-MS	polyphenolics hydroxycinnamate and flavonoids quercetin-3-*O*-(2″-*O*-apiosyl-6″-*O*-rhamnosyl)glucoside-7-*O*-glucoside, kaempferol-3-*O*-(2″-*O*-apiosyl-6″-*O*-rhamnosyl)glucoside, quercetin-3-*O*-glucoside, quercetin-3-*O*-rhamnoside; and K3R, kaempferol-3-*O*-rhamnoside	[203]
* **Momordica charantia** *				
40.	*Momordica charantia* fruit (Cucurbitaceae)	LC-MS-QTOF	brevifolincarboxylic acid (new antioxidant molecule), 3-malonylmomordicin I and goyaglycoside G and other metabolites	[204]
**Cabbage**				
41.	Chinese cabbage—root colonization by *Piriformospora indica*	High-throughput GC-MS	total compounds—1126, 549 identified in colonized and non-colonized cabbage roots and hyphae of *P. indica*	[205]
42.	Cabbage (*Brassica oleracea var. capitata*)—taste attributes	GC-MS	primary metabolites, 4-aminobutyric acid, fructose 1-phosphate, adipic acid, 5-oxoproline, *N*-acetylglycine, *O*-phosphoethanolamine, and homovanillic acid	[206]
43.	*Brassica rapa* subsp. chinensis	GC-TOF-MS and HPLC	53 primary metabolites in 3 pak choi cultivars, 49 hydrophilic metabolites in others.	[207]
**Spinach** (*Spinacia oleracea*)				
44.	Leaves	GC-MS	variations in carbohydrates, organic acids, amino acids content after treatment with NO^−^_3_ and Gly	[208]
**Common beans** (*Phaseolus vulgaris* L.)				
45.	Contrasting genotypes of saline conditions	GC-MS	accumulation of lysine, valine and isoleucine in roots	[209]
46.	Common bean Genotype (107 accessions)-heat stress	Non-targeted and targeted MS	measured high content of salicylic acid while low content of triterpene saponins, phenylpropanoids deciphered	[210]
47.	Cold-tolerant (120) and sensitive (93) common bean cultivars	UPLC-MS	flavonoids, methionine and malondialdehyde as biomarkers against cold temperatures	[211]
**Lotus** (*Nelumbo nucifera*)				
48.	Seeds	LC-MS	identified flavonoids (30) and alkaloids	[212]
49.	Seeds	LC-MS	amino acids, organic acids, sugars	[213]
**Moringa** (*Moringa oleifera*)				
50.	Leaves and stem	^1^H-NMR	30 metabolites identified, 22 common in leaf and stem tissues. 4-aminobutyrate, adenosine, guanosine, tyrosine, *p*-cresol	[214]

LC—liquid chromatography, MS—mass spectrometry, NMR—nuclear magnetic resonance, QTOF—quadrupole time-of-flight, HRMAS NMR—high-resolution magic-angle-spinning nuclear magnetic resonance, PCA—Principal component analysis; PLS-DS—partial least square discriminant analysis, qNMR—quantitative NMR, UPLC—Ultra-Performance Liquid Chromatography, MWDB—MetWare database, MRM—multiple reaction monitoring.

## 5. Conclusions

Metabolomics has become a versatile tool to investigate the metabolic status of vegetable plants with multi-faceted dimensions. These crops are being grown by farmers worldwide, not only for the nutritional and pharmaco-physiological prospective, but also for income generation and livelihood for resource poor rural people including youth and women. So, crop plants with high nutrition, enhanced productivity and low pest/disease resurgence could be of much demand by farmers for gaining improved economic viability from farms. However, there remain multifarious interactions of vegetable plants with their environment and biotic entities that influence quality, nutritional richness and productivity of the crops. For similar reasons, the importance of metabolic composition, i.e., the metabolome of the crop produce under specified agroclimatic regimes and its implications on human health becomes prominent. To measure metabolome is to measure the ultimate functional chemical entities of the cells that firstly and directly respond to the challenges of the environment. However, extracting and identifying all the metabolite molecules from plant cells and tissues is not realistic and one can extract only maximum possible metabolites by efficient solvent combinations and extraction conditions. Metabolomics offers limitless opportunities without analysis, application and interpretation related limitations to decipher global as well as targeted metabolites from vegetable plants. The area is ever-evolving and holds promise for the developers of the methods, techniques, protocols, algorithms and statistical tools. 

Vegetable plant interactions with beneficial microorganisms, pathogens, pests, environment, edaphic conditions and anthropogenic factors invoke such metabolic changes which, if marked at global scale, could yield biomarker candidate metabolites for utilization in crop improvement through identified mQTLs. Knowing in planta candidate metabolites that act as phytohormones, signal molecules, cross-communication agents, elicitors, surface-inhibitors, antimicrobials, and stimulators/suppressors of metabolic pathways, synergizing agents of metabolism and post-harvest signatures, specific metabolic pathways could be tuned in favor of plant health. Mining of crop plant VOCs using GC-MS metabolomics platform is an effective way to identify molecules that could act as candidates for biosensor development. The rhizosphere of crops is the hotspot for diverse microbial communities and their multi-functionalities. Modulating overall beneficial microbiome functions in the rhizosphere, either through exogenous microbial inoculation or by the application of growth-stimulating organic compounds, is a strategy to provoke better crop performance that can be studied for candidate metabolic alterations using metabolomics. Growth stage-dependent crop metabolome pool in plants under conventional crop production practices (that involve synthetic chemical fertilizers and pesticides), integrated pest management (IPM), integrated nutrient management (INM), low-external input farming, natural methods and organic cultivation can be mapped effectively using metabolomics. Identifying metabolites from the plants growing in different farming practices and implying them in molecular breeding programs for improved nutrition, resistance and crop health are new and promising avenues for future agriculture. Integrated metabolomics platforms entwined with the genomic, transcriptomic and proteomic resources could help to explore key metabolites, crucial metabolic pathways and critical networks engaged by the plants to overcome stress-linked challenges in vegetable crops. 

## Figures and Tables

**Figure 1 ijms-23-12062-f001:**
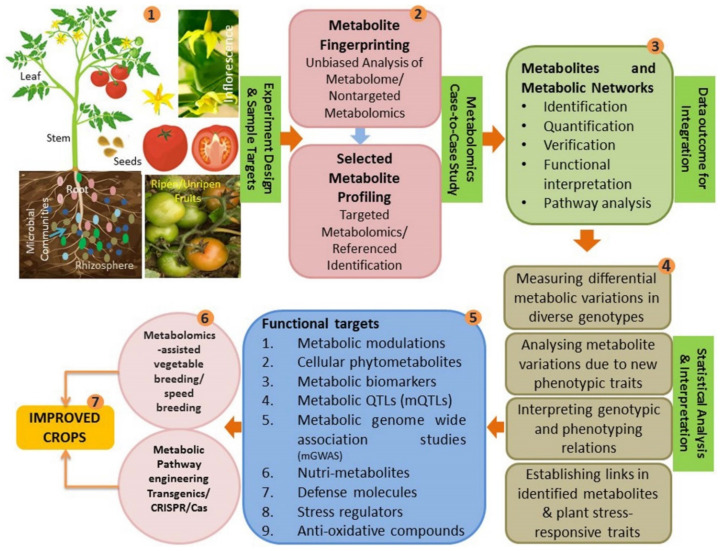
Schematic presentation of metabolomics applications in crop improvement programs: 1. A representative sampling source of vegetable crop plants (tomato) as a biological source from which cellular metabolome can be extracted from almost all the plant parts and the rhizosphere under varying experimental environmental conditions; 2. data acquisition approach in metabolomics to be applied whether unbiased non-targeted fingerprinting is required or the analysis and quantification of a few selected target molecules is the need of the experiment; 3. study outcomes which needs biological interpretation for hypothesis questions; 4. possible answers to the hypothesis questions in the cellular chemistry and its entwining relations with the environmental impacts; 5. functional targets that could be achieved through metabolomics analysis of the vegetable plants; 6. result-oriented applications of the data outcomes in crop improvement practices; and 7. the ‘end product’ of the experimental metabolomics exercise in vegetable crops.

**Figure 2 ijms-23-12062-f002:**
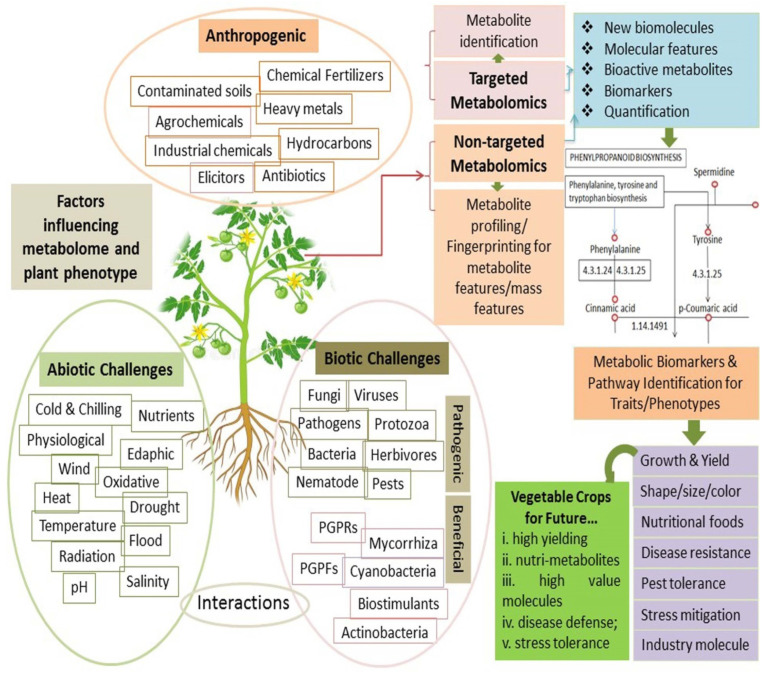
Illustrative diagram of possible plant environmental interactions, which are supposed to influence the metabolic status of the crop plants. Analyzing the metabolome of plants exposed to such challenges using metabolomics approaches can yield competitive vegetables crops with better yield, high level of defense and stress mitigating capabilities.

**Figure 3 ijms-23-12062-f003:**
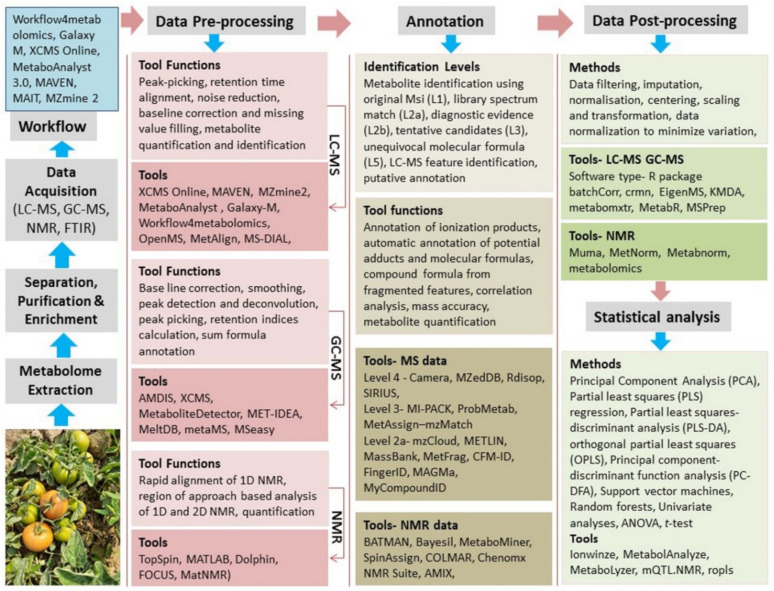
Illustrative flow diagram of metabolomics methods, techniques and step-wise bioinformatics resources for identifying metabolites and deciphering their role in the metabolic pathways.

## Data Availability

Not Applicable.

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
