# Peer review of "Metabolomics-Driven Mining of Metabolite Resources: Applications and Prospects for Improving Vegetable Crops"

_ijms, 2022, doi:10.3390/ijms232012062_

Round 1
Reviewer 1 Report
Singh et al. reported Application Perspectives of Metabolomics for Mining Metabolic Resources in Vegetable Crops, in which, the authors discussed (1) vegetable plants as prominent metabolite resources, (2) metabolites, metabolome and metabolomics, (3) application perspectives of metabolomics in vegetable crops, (4) crop domestication studies, (5) monitoring development-dependent metabolic changes, (6) nutritional metabolomics of vegetable plants, (7) decepheing plant’s adaptive mechanisms and mitigation strategies under abiotic stresses, (8) assessing metabolome under plant-microbe interactions, (9) metabolomic studies in plant-insect (herbivore) interactions, (10) understanding on rhizosphere metabolome, (11) metabolomic studies on vegetable transenic plants, (12) metabolomic changes in vegetable plants due to nanoparticles impact, and (13) metabolome profiling and pharmaco-physiological impact assessment. This is an extensive review and carries good information which could interest the readership of International Journal of Molecular Sciences. However, there are extensive literature cited and less in-depth insights from the authors in this review. In addition, according to the content of the MS, the title of the review is not precise. The authors should point out what are the perspectives, challenges or disadvantages of metabolomics used for mining metabolome in vegetable crops.
I recommend major revision.
Below are some comments and edits to improve the content
Line 171 5. Crop domestication studies, I recommend as vegetable crop domestication studies.
Line 181–182, “using GC-MS……in fruits and leaves”, not italic.
Line 193–194, what indicates when cited reference 87, this reference only summarized the preparatory steps for metabolite profiling of polar compounds by GC-MS in tomato fruit. I think it is out of the scope of monitoring development-dependent metabolic changes. Please reassess the relevance of all references to each topic.
Line 294, Comparatiove, please check.
Line 338, I am confused about the section 10 and 11, in the section 10, I think, the authors tried to discuss the metabolomics studies in plant-insect (herbibore) interaction, however, the authors mentioned rhizosphere metabolomics in the third paragraph, which was reported in the section 11, understanding on rhizosphere metabolome. Why did not the authors combine them?
Line 348, are you sure that the authors of the reference 153 used headspace liquid chromatography SPME for revealing volatile compounds? The original text: We performed widely targeted comparative metabolome and volatilome profiling by ultraperformance liquid chromatography–tandem mass spectrometry (UPLC-MS/MS) and headspace solid-phase microextraction coupled to gas chromatography–mass spectrometry (HS-SPME/GC-MS), respectively, on eggplants and tomatoes under control and T. absoluta infestation conditions.
Line 369–371, period (.) may be lost.
Line 385–386, “in the soil……”, not italic.
Line 403, it should be transgenic, not transenic.
Line 467, “Ientifying”, Identifying.
Please use the Oxford comma.
Please check the format of all references in the MS.
The quality of figures should be improved, especially fig. 2.
Please check Table 1 carefully, for example, gamma-aminobutyrate (GABA) and γ-amino-butyric acid (GABA), β-Ioneone in Sr. No. 3, lignin structure in Sr. No. 6, etc. Anyway, the format of the table is not consistent.
Author Response
Authors thank the Reviewer 1 for suggesting meaningful and critical revision to the MS. Following the suggestions, we have now looked at every corner of the MS and modified it by significant additions or deletions at many places, which well be reflected the text.
I hope the revisions have significantly improved the MS for successful publication.
Point-wise Revision based on Reviewer 1 suggestions is attached.

Reviewer 2 Report
Line 62: Here you are discussing “Vegetable plants as prominent metabolite resources” but the section lacks specific example of crops. I would add vegetable plant examples for making suitable arguments.
Line 101: rather than using the heading “3. Metabolites, metabolome and metabolomics”- change it to the goal of that section. Metabolites, metabolome and metabolomics means nothing, and reader can’t comprehend anything before reading.
Line 160: Section “Application perspectives of metabolomics in vegetable crops” needs expanding. I think some GC-MS and LC-MS examples can be a good addition for this section. Again, be very specific when giving examples with plants species as reader wants to know what plant reports are present.
Line 171: “Crop domestication studies” title sounds mundane. Crop domestication studies reveal what? A brief would be your title. This section is actually written very well with proper examples but limited to tomato. Please expand with more examples.
Line 270: “8. Decepheing plant’s adaptive mechanisms and mitigation strategies under abiotic stresses”. Typo ‘Deciphering’. The end of the paragraph should a general statement on what was deciphered.
Figure 1 has problems on texts formatting and fonts.
Figure 2 from quantification to pathway step. The pathways are absolutely not readable, and it needs to have better resolution, otherwise the figure becomes incomplete.
-Overall, I think the review has a lot of content, but it was just summary of whatever other people have done in this field. Authors didn’t make any interpretations based on previous studies which is the key for writing a review paper. Every section should have this otherwise the review fails to predict directions.
Author Response
Pointwise revisions undertaken following Reviewer 2 suggestions is attached.

Round 2
Reviewer 1 Report
The authors have improved the manuscript and I recommend accepting in its present form.